# Pediatric COVID-19: clinical characteristics and prognostic outcomes

Jing Chen,[1] Caijin Yan,[2] Yuling Pan,[3] Mei Yang,[2] Binli Zhong,[4] Gangxi Lin,[5,6] Yafang Hong[7,8]

**ABSTRACT**    To analyze the clinical characteristics of pediatric coronavirus disease 2019 (COVID-19) and provide direction for clinical treatment and nursing. Children under 18 years old admitted to the First Affiliated Hospital of Xiamen University were collected due to testing positive for COVID-19 from the outbreak of COVID-19 to January 2023. Patient data, including age, vaccination history, admission symptoms, clinical progression, and treatment outcomes, were extracted from the database. From April 2020 to January 2023, 237 children under 18 years old with COVID-19-positive result in the pediatric ward were collected. All cases achieved hospital discharge without mortality. Among them, 17.72% were under 1 year old, 27.00% were 1–4 years old, 34.60% were 5–10 years old, and 20.68% were over 10 years old. Male predominance was observed (63.29%, sex ratio 1.72:1), the prevalence rate of children of all ages is also similar in sexual distinction ($P = 0.739$). Additionally, 53.16% of patients have mild symptoms; the proportion of asymptomatic COVID-19 positive children over 10 years old is higher than that of other age groups. The proportion of respiratory symptoms of children under 1 year old is significantly higher than other age groups (88.09%, $P < 0.01$). Convulsions manifested universally in 23 confirmed pediatric COVID-19 cases (aged 2 months–11 years). Pediatric COVID-19 infections generally manifest mild symptoms; however, infants under 1 year old exhibit more severe respiratory symptoms, necessitating escalated ventilatory support. Additionally, male children demonstrate higher susceptibility to infection compared with females across all pediatric age groups, carrying clinical and public health implications for epidemic response.

**IMPORTANCE** According to a review of 72,314 cases by the China Center for Disease Control and Prevention (CDC), less than 1% of the cases occurred in children under the age of 10 years, with the vast majority being children. If children are indeed infected with severe acute respiratory syndrome coronavirus type 2 (SARS-CoV-2), the vast majority of people will have mild diseases that do not require hospitalization. Therefore, the available data on coronavirus disease 2019 (COVID-19) in infants are still limited. This article collects the cases of children who have been hospitalized because of positive tests since the outbreak of COVID-19, analyzes, and understands how COVID-19 influences children, which can help us better interpret and respond to public health measures.

**KEYWORDS**    SARS-CoV-2, COVID-19, children, infant, mild, convulsions

I n December 2019, the coronavirus disease 2019 (COVID-19) caused by the new severe acute respiratory syndrome coronavirus type 2 (SARS-CoV-2) broke out for the first time in Wuhan, China. Since then, COVID-19 has overwhelmed medical systems around the world, which influences all age groups and has been the main cause of global mortality and incidence rate. However, the incidence rate and mortality rates of COVID-19 vary significantly across gender and age differences. Compared with adults, SARS-CoV-2 causes milder symptoms and fewer deaths in children and adolescents.

**Peer Reviewer** Kimiya Kazemi Esfeh, Ahvaz Jondishapour University of Medical Sciences, Ahvaz, Iran

Address correspondence to Binli Zhong, 89344969@qq.com, Gangxi Lin, lingangxi999@sina.com, or Yafang Hong, 928485009@qq.com.

Jing Chen, Caijin Yan, and Yuling Pan contributed equally to this article. Author order was determined on the basis of seniority.

The authors have no conflicts of interest.

Nevertheless, they are still susceptible to SARS-CoV-2 infection and may transmit the virus to others. From the outbreak of COVID-19 to October 2021, the incidence rate of children under 5 years old is 2% (1,890,756), accounting for 0.1% (1,797) of the global reported deaths. Older children and younger adolescents (5 to 14 years old) account for 7% of global reported cases (7,058,748) and 0.1% of global reported deaths (13,28) (1).

According to a review of 72,314 cases by the China Center for Disease Control and Prevention (CDC), less than 1% of the cases occurred in children under the age of 10 years, with the vast majority being children. If children are indeed infected with SARS-CoV-2, the vast majority of people will have mild diseases that do not require hospitalization. Therefore, the available data on COVID-19 in infants are still limited. This article collects the cases of children who have been hospitalized because of positive tests since the outbreak of COVID-19, analyzes, and understands how COVID-19 influences children, which can help us better interpret and respond to public health measures (2).

## MATERIALS AND METHODS

This is a retrospective study. We collected children younger than 18 years old who were admitted to the First Affiliated Hospital of Xiamen University, which is the designated hospital for COVID-19 in Xiamen, due to testing positive for COVID-19 from the outbreak of COVID-19 to January 2023. A total of 237 cases of COVID-19 positive children younger than 18 years old with a confirmed diagnosis of COVID-19 with PCR in our hospital were included in this study. Clinical characteristic information about the patient from the database, including age, history of SARS-CoV-2 vaccination, signs and symptoms upon admission, clinical course, and treatment outcomes, was collected. This study mainly analyzes the impact of age, gender, and vaccination on the severity of COVID-19 in children. Patients were divided into four groups according to age: infant group: young children between the ages of 1 and 12 months, toddler group: children between the ages of 1 and 4 years old, child group: children between the ages of 5 and 9 years old, younger adolescents: adolescents between the ages of 10 and 17 years old. Inclusion criteria: children under 18 years old with confirmed COVID-19 admitted to the First Affiliated Hospital of Xiamen University. Exclusion criteria: age greater than 18 years. Discharge criteria: two consecutive negative PCR nucleic acid tests (CT value >35).

### Diagnostic criteria

Suspected cases meeting any one of the following criteria are considered confirmed cases (3, 4):

1. Positive SARS-CoV-2 test result by real-time reverse transcription polymerase chain reaction (RT-PCR).

2. High sequence homology between genetic material from respiratory tract or blood samples and known SARS-CoV-2.

3. Concurrent positivity for SARS-CoV-2-specific IgM and IgG antibodies in serum.

4. Seroconversion (IgG changing from negative to positive) or a ≥4-fold increase in SARS-CoV-2-specific IgG antibody levels in convalescent-phase serum compared with acute-phase levels.

### Clinical classification as below

Asymptomatic: the respiratory tract and other specimens tested positive for SARS-CoV-2 etiology, and there were no related clinical manifestations in the whole infection process (3).

Mild cases are mainly characterized by acute upper respiratory tract infection. Clinically, the presentation and physical examination showed no signs of lower respiratory tract involvement.

Medium: type has respiratory symptoms, the respiratory rate (respiratory rate, RR) is <30 times/min, and the oxygen saturation is >93%when breathing air in the resting state.

Imaging examination showed pneumonia changes, but did not reach the level of severe pneumonia.

Heavy: meets any one of the following items. (i) Shortness of breath occurs:<2 months old, RR ≥ 60 times/min; 2 to 12 months old, RR ≥ 50 times/min; 1 to 5 years old, RR ≥ 40 times/min;>5 years old, RR ≥ 30 times/min, excluding the effects of fever and crying. (ii) In the resting state, when breathing air, the oxygen saturation is ≤93%. (iii) Dyspnea: accompanied by moaning, nasal flaring or three-concave sign, cyanosis, and intermittent apnea. (iv) Pulmonary imaging examination results show bilateral or multiple lung lobes infiltration, rapid progression of lesions >50% in a short period of time, or pleural effusion.

## Statistical analysis

SPSS software was performed for statistical analysis (IBM, Armonk, New York, USA). Kruskal-Wallis H test was used in the comparison of numerical variables when appropriate. $P$-values < 0.05 were established as the level of significance.

## RESULTS

A total of 237 cases of COVID-19-positive children younger than 18 years old with a confirmed diagnosis of COVID-19 with PCR in our hospital were included in this study. All of them were ultimately discharged from the hospital, with no deaths reported. Accounts for 17.72% ($n$ = 42) were infants (<1 year old), 27.00% ($n$ = 64) were young children (1–4 years old), 34.60% ($n$ = 82) were children aged 5–10 years old, and children over 10 years old accounted for 20.68% ($n$ = 49). More than half of the affected children are male 63.29% ($n$ = 150), with the sex ratio of 1.72 (male/female), and the prevalence rate of children of all ages is also similar in sexual distinction ($P$ = 0.739). In total, 68.78% (163) were not vaccinated, and 31.22% (74) were partially or completely vaccinated. Among them, 11 cases (4.64%) were asymptomatic infections, 53.16% (126/237) of patients had mild symptoms, 99 cases (41.77%) had moderate symptoms, and only one case had severe symptoms. Two children developed multisystem inflammatory syndrome in children (MIS-C), five cases had myocardial damage, and three children developed viral encephalitis. Ten children experienced symptoms of muscle weakness, with two patients experiencing only muscle weakness, ranging in age from 4 to 12 years old. The results of the characteristics of children in different groups are shown in Table 1.

The main symptoms of COVID-19 include gastrointestinal tract (25/237, 10.55%), respiratory system (154/237, 64.98%), fever (171/237, 72.15%), and convulsions (23/237, 9.70%), of which 7.17% (17/237) require mechanical ventilation. The proportion of asymptomatic COVID-19-positive children over 10 years old is higher than that of other age groups. The proportion of respiratory symptoms of children under 1 year old is significantly higher than other age groups (88.09%, $P$ < 0.01). At the same time, the proportion of patients requiring auxiliary ventilation is also significantly higher (19.05%, $P$ < 0.01). More than half of children under 1 year old have abnormal lung CT (61.90%, $P$ = 0.03), and the proportion of moderate symptoms is also significantly higher than that of other age groups (64.28%, $P$ < 0.01) with no severe symptoms.

A total of 23 children with positive COVID-19 had convulsions, and the age ranged from 2 months to 11 years old, 65.22% of whom were male. The average age is 4.53 years old, with only one infant. All children with convulsions have symptoms of fever, and the average length of hospital stay is 5.39 days. More than half had moderate to severe symptoms, more than 60% had abnormal lung CT, and most of them were not vaccinated.

## DISCUSSION

Since the outbreak of COVID-19, researchers have gradually found that there are significant age differences in incidence rate and mortality of COVID-19. SARS-CoV-2 can cause many diseases in adults, while in children, its incidence and severity of the disease significantly decrease (5). Our data are similar, the incidence rate of COVID-19

**TABLE 1** Characteristics of COVID-19 in different age groups children (n [%])

| Characteristics | Infant | Toddler | Child | Younger adolescent | P-value |
|---|---|---|---|---|---|
| N | 42 | 64 | 82 | 49 | |
| Male | 28 (66.67) | 37 (57.81) | 54 (65.85) | 31 (63.27) | 0.739 |
| Female | 14 (33.33) | 27 (42.19) | 28 (34.15) | 18 (36.73) | |
| Fundamental disease | | | | | 0.388 |
| Yes | 39 (92.86) | 53 (82.81) | 67 (81.71) | 40 (81.63) | |
| No | 3 (7.14) | 11 (17.19) | 15 (18.29) | 9 (18.37) | |
| Vaccination of COVID-19 | | | | | 0.000 |
| No vaccination history | 42 (100) | 60 (93.75) | 41 (50) | 20 (40.82) | |
| One dose of vaccination | 0 | 0 | 4 (4.88) | 2 (4.08) | |
| Two doses of vaccination | 0 | 4 (6.25) | 37 (45.12) | 27 (55.10) | |
| Combined symptoms | | | | | 0.085 |
| Symptom-free | 0 | 2 (3.13) | 5 (6.10) | 4 (8.16) | |
| Less than two symptoms | 23 (54.76) | 37 (57.81) | 49 (59.76) | 29 (59.18) | |
| Three or more symptoms | 19 (45.24) | 25 (39.06) | 28 (34.15) | 16 (32.65) | |
| Symptomatic | | | | | |
| Fever | 34 (80.95) | 54 (84.38) | 53 (64.63) | 30 (61.22) | 0.009 |
| Convulsions | 1 (2.38) | 12 (18.75) | 8 (9.76) | 2 (4.08) | 0.016 |
| Respiratory system | 37 (88.09) | 42 (65.62) | 51 (62.20) | 24 (48.98) | 0.001 |
| Gastrointestinal tract | 4 (9.52) | 9 (14.06) | 9 (10.98) | 3 (6.12) | 0.591 |
| Classification | | | | | 0.005 |
| Asymptomatic | 0 | 2 (3.12) | 5 (6.10) | 4 (8.16) | |
| Mild | 15 (35.71) | 42 (65.63) | 47 (57.32) | 22 (44.90) | |
| Medium | 27 (64.28) | 19 (29.69) | 30 (36.59) | 23 (46.94) | |
| Severe | 0 | 1 (1.56) | 0 | 0 | |
| Days in hospital | 6.98 ± 3.57 | 9.81 ± 8.34 | 11.93 ± 8.61 | 14.06 ± 10.71 | 0.000 |
| Assisted ventilation | 8 (19.05) | 1 (1.56) | 5 (6.10) | 0 | 0.005 |
| Abnormal lung CT | 26 (61.90) | 24 (37.5) | 30 (36.59) | 23 (46.94) | 0.036 |
| MIS-C | 0 | 1 | 1 | 0 | |
| Myocardial damage | 3 | 2 | 0 | 0 | |
| Acute laryngitis | 8 | 2 | 4 | 0 | |
| Viral encephalitis | 1 | 1 | 0 | 1 | |

in infants (<1 year old) is the lowest (17.72%), and SARS-CoV-2 infection in children is mainly characterized by mild to moderate symptoms, with over half of the children (53.16%) experiencing mild symptoms. Adult infection with SARS-CoV-2 is a highly inflammatory response characterized by acute respiratory distress syndrome (ARDS) as the main symptom (6), but symptoms are mild in children (2, 7). The most likely reason is that there is an age difference in the expression of angiotensin-converting enzyme 2 (ACE2). The receptor for SARS-CoV-2 spike (S) glycoprotein binds to the cell surface through the ACE2 membrane, initiates S protein activation through the transmembrane protease serine 2 (TMPRSS2) binding to ACE2, and mediates the fusion of the virus and cell membrane, which is a key determinant of SARS-CoV transmission. After entering the host cell, SARS-CoV-2 downregulates the expression of ACE2 on the cell surface, thereby minimizing the enzyme's ability to exert protective effects on organs (8). The function of ACE2 is to convert angiotensin II into its metabolite angiotensin. Reduced expression of ACE2 can lead to chronic heart failure and lung damage; therefore, ACE2 imbalance in COVID-19 may disrupt the balance of angiotensin II/angiotensin, leading to inflammation and hypoxia (9). ACE2 is widely expressed in the heart, lungs, and gastrointestinal systems, but the highest expression of ACE2/TMPRSS2 is observed in nasal epithelial cells (10), which also explains the symptoms of olfactory loss in adults infected with SARS-CoV-2. However, the expression of ACE2 in the nasal epithelium of children is relatively low (10), which reasonably explains the low infection rate and mild

symptoms in children. Another possibility is that the virus has evolved from the Delta strain to the Omicron variant. Numerous studies have demonstrated that the Omicron variant tends to result in milder COVID-19 symptoms and a shorter clinical course in patients, which may be associated with its reduced cytotoxicity (11–13).

More than half of the COVID-19 positive children in our data are male 63.29% ($n$ = 150), with a gender ratio of 1.72 (male/female). The incidence rates among children of all ages are also similar in terms of gender differences ($P$ = 0.739). In an Italian statistical data, the mortality rate between men and women is 1.75 (male/female) (14). In the data of confirmed COVID-19 cases from multiple countries, it was also found that the proportion of male patients hospitalized for COVID-19 treatment is 50% higher than that of female patients, and the number of patients admitted to the ICU is 2–3 times higher than that of female patients (15, 16). It can be seen that males have a higher susceptibility compared with females, and there is no correlation with age. We can explain it through the following aspects: First, male ACE2 is more expressed in tissues than female ACE2 (17). Fischer et al. found ovariectomy leads to an increase in ACE2 activity in females, while orchiectomy in males reduces ACE2 activity. In addition, when ovarian tissue is removed in women, the expression of ACE2 increases in their kidneys and adipose tissue and decreases in subsequent estrogen replacement therapy (18). Female mice subjected to ovariectomy or treated with estrogen receptor antagonists exhibited increased mortality (19). From this, we can infer that sex hormones may affect the expression and activity of ACE2, thereby influencing the ability of SARS-CoV-2 to enter cells by mediating viral attachment to target cells (14, 20). Channappanavar et al. noted that male mice were more susceptible to SARS-CoV than age-matched female mice, and this increased susceptibility was associated with higher viral titers, enhanced vascular leakage, and alveolar edema (19). Second, androgen receptors can regulate the transcription of TMPRSS2 (21), and the expression of TMPRSS2 in males is higher than that in females, especially in bronchial epithelial cells (14), promoting the fusion of SARS-CoV-2 with cell membranes and invading cells. Finally, males have a lower innate recognition and response to viruses compared with females. Females have a higher number and activity of innate immune cells, which can generate stronger immune responses. That is why the infection rate and intensity of the virus are usually lower in women (22, 23).

In our data, 10 children experienced symptoms of muscle weakness, with two patients experiencing only muscle weakness, ranging in age from 4 to 12 years old. Five children experienced myocardial injury. There are also many literature reports on the occurrence of complications, such as muscle weakness in COVID-19 (24), and even some patients may experience rhabdomyolysis (25). This may be due to COVID-19 infection, which leads to an increase in energy demand, followed by pneumonia and respiratory symptoms causing hypoxemia, exacerbating insufficient oxygen supply and causing a continued decrease in energy supply (26). Damaged energy supply can cause oxidative stress, induce endothelial and mitochondrial dysfunction, and further weaken energy supply (27). In addition, there were 23 COVID-19 positive children with convulsions, and the age ranged from 2 months to 11 years old. Although these children with seizures are accompanied by fever symptoms, they exceed the typical age range of febrile seizures, which is less than 6 months and more than 5 years old. Febrile convulsion refers to the convulsion caused by fever or the discovery of epilepsy, and there is no central nervous system infection (28). It can be inferred that the convulsion caused by COVID-19 is not exactly the same as that caused by febrile convulsion. Even during the COVID-19 pandemic, the incidence rate of febrile convulsions decreased by 36% (29). The emergence of COVID-19 was accompanied by a significant decline in other respiratory viruses, such as influenza, largely attributable to public health interventions (30). The decrease in the incidence rate of febrile convulsions may be caused by the respiratory infectious diseases in the incidence rate of influenza, chickenpox, and other infectious diseases known to lead to febrile convulsions due to social distance, wearing masks, and hand hygiene (31). SARS-CoV-2 and influenza viruses share similar transmission routes—

via respiratory droplets and contact—so measures that reduce or halt the spread of COVID-19 should exert comparable effects on influenza (32). Chow et al. argue that evolution is pivotal to immune evasion in respiratory viruses, with genetic mutations across multiple viral species expanding their gene pools. The abrupt decline in respiratory virus transmission may have created an evolutionary bottleneck, thereby impacting the genetic diversity of numerous viral species (33). Meltzer et al. found that when vitamin D is insufficient, the risk of COVID-19 infection is significantly increased (34). Vitamin D also can affect the metabolism of zinc, which can reduce the replication of coronavirus (35). There are also studies that have found a negative correlation between vitamin D and interleukin 6, which is a key cytokine in inflammatory storm induced by COVID-19 infection (36). While biological and technical variables may impact the commutability of cytokine immunoassays and complicate the interpretation of their results, an IL-6 receptor blocker has been approved based on multicenter, randomized controlled trials for treating COVID-19 pneumonia patients with elevated IL-6 levels (37). In addition, COVID-19 patients also found a significant decrease in serum Ca2+ (38), indicating that during the COVID-19 pandemic, SARS-CoV-2 may participate in the occurrence of seizures by interfering with vitamin calcium metabolism.

The proportion of asymptomatic children over 10 years old with positive COVID-19 in our data is higher than other age groups. However, infants (under the age of 1 year old) experiencing respiratory symptoms were significantly higher than other age groups (88.09%, $P < 0.01$) and also had a higher proportion of requiring assisted ventilation (19.05%, $P < 0.01$). The proportion of moderate symptoms in infants was also significantly higher than other age groups without severe symptoms (64.28%, $P < 0.01$). Although children infected with COVID-19 usually have milder symptoms compared with adult patients, the severity is higher for younger individuals under the age of 1 year old. An epidemiological data from China also found that young children, especially infants, are susceptible to SARS-CoV-2 infection (39). Young children are more susceptible to respiratory virus infections in winter and are often co-infected, which may lead to overlapping symptoms and signs, resulting in more severe respiratory symptoms in children under 1 year old (40). With the lifting of quarantine measures, in addition to COVID-19, other common respiratory viruses have also rebounded. Research has found that the virus co-detection rate in specimens treated at hospitals is 10%–50% (41, 42). In addition, maternal antibodies can protect newborns from various infections, but their protection time is limited and gradually decreases from birth or cessation of breastfeeding, usually lasting for about a year (43). However, the study is based on data from a single center, and because of China's special lockdown policy, there are fewer children with positive COVID-19 during the lockdown period. After the release on 20 December 2022, compulsory COVID-19 testing will no longer be carried out, resulting in the limited number of positive COVID-19 cases we collected. Owing to the limitations of the data, the analysis of results and their extrapolation may be impacted. From this limited amount of data, we aimed to identify some patterns regarding children's COVID-19 infections, so as to address the anxiety of the patients' families and develop effective coping measures for nursing care.

## Conclusion

Although most symptoms of COVID-19 infection in children are mild, it does not mean that we do not need to take it seriously. Especially for children under the age of 1 year old, respiratory symptoms are more pronounced and are more prone to respiratory support, requiring special attention. Second, male children seem to be more susceptible to COVID-19 infection than females, and this is similar in children of all ages. At present, although the data we provide are limited, this information may provide reference value for clinical doctors and the public in responding to the ongoing pandemic. Further research is needed to clarify the mechanism of COVID-19 in children and provide more directions for future treatment.

## ACKNOWLEDGMENTS

We thank all patients involved in the study.

This work was Supported by Fujian Provincial Department of Science and Technology, Youth Innovation Program of the Natural Science Foundation (2023J05276).

Y.H., B.Z., and G.L. conceived and designed the study. C.Y., Y.P., and M.Y. collected the epidemiological and clinical data. J.C. and C.Y. processed statistical data. J.C. drafted the manuscript. Y.H., B.Z. and G.L. revised the final manuscript. All the authors approved the final version of the manuscript.

## AUTHOR AFFILIATIONS

[1]Department of Reproductive Medicine, The First Affiliated Hospital of Xiamen University, School of Medicine, Xiamen University, Xiamen, People Republic of China
[2]Department of Pediatrics, the First Affiliated Hospital of Xiamen University, Xiamen, People Republic of China
[3]Department of Gynecology and Obstetrics, the First Affiliated Hospital of Xiamen University, Xiamen, People Republic of China
[4]Department of General Medicine, the First Affiliated Hospital of Xiamen University, Xiamen, China
[5]The School of Clinical Medicine, Fujian Medical University, Fuzhou, People Republic of China
[6]Department of Pediatric, the First Affiliated Hospital of Xiamen University, School of Medicine, Xiamen University, Xiamen, People Republic of China
[7]Department of Hospital Infection Control, First Hospital of Quanzhou Affiliated to Fujian Medical University, Quanzhou, People Republic of China
[8]School of Public Health, Xiamen University, Xiamen, People Republic of China

## AUTHOR ORCIDs

Jing Chen  http://orcid.org/0000-0001-8923-9971
Binli Zhong  http://orcid.org/0009-0002-8024-2183
Gangxi Lin  http://orcid.org/0009-0006-4791-7045
Yafang Hong  http://orcid.org/0009-0006-9140-8360

## DATA AVAILABILITY

The study has been clinically filed in the Medical Research Registration and Filing Information System (no. MR-35-24-021747, https://www.medicalresearch.org.cn.). The data that support the results of this study are available from the corresponding author upon reasonable request.

## ETHICS APPROVAL

According to the declaration of Helsinki, this study was approved by the Ethics Committee of the First Affiliated Hospital of Xiamen University (no. SL-2024KY084-01). Written informed consent for participation was not required for this study.

## ADDITIONAL FILES

The following material is available online.

Open Peer Review

**PEER REVIEW HISTORY (review-history.pdf).** An accounting of the reviewer comments and feedback.

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
