## [Reviewer comments · Microbiology Spectrum]

Microbiology Spectrum

Pediatric COVID-19: Clinical characteristics and prognostic outcomes

Jing Chen, Caijin Yan, Yuling Pan, Mei Yang, Binli Zhong, Gangxi Lin, and Yafang Hong

Corresponding Author(s): Yafang Hong, Fujian Medical University Affiliated First Quanzhou Hospital

Review Timeline:

Submission Date:	May 21, 2025
Editorial Decision:	June 28, 2025
Revision Received:	July 19, 2025
Accepted:	July 24, 2025

Editor: Benjamin Liu

Reviewer(s): Disclosure of reviewer identity is with reference to reviewer comments included in decision letter(s). The following individuals involved in review of your submission have agreed to reveal their identity: kimiya kazemi esfeh (Reviewer #1)

Transaction Report:

DOI: <https://doi.org/10.1128/spectrum.01577-25>

Re: Spectrum01577-25 (**Clinical characteristics and outcomes of COVID-19 for children**)

Dear Ms. yafang Hong:

Thank you for the privilege of reviewing your work. Below you will find my comments, instructions from the Spectrum editorial office, and the reviewer comments.

Editor's comments:

1. Omicron with key mutations in envelope gene is the direct reason of the mild clinical characteristics and outcomes of COVID. The authors should introduce or discuss this point. More references should be cited, with this one (PMID: 39744807) as an example (citing is optional).
2. Introduction should include diagnostic methods for COVID. More references should be cited, with this one (PMID: 39857007) as an example (citing is optional).
3. "Adult infection with SARS-CoV-2 is a highly inflammatory response characterized by acute respiratory distress syndrome (ARDS) as the main symptom, but symptoms are mild in children": There are no references to support this statement. More references should be cited, with this one (PMID: 33337932) as an example (citing is optional).
4. "The decrease in the incidence rate of febrile convulsions may be caused by the decrease in the incidence rate of influenza, chickenpox and other infectious diseases known to lead to febrile convulsions due to social distance, wearing masks and hand hygiene". The authors discuss the interplay between COVID and flu. But COVID also has influence on flu epidemiology. The authors should discuss this. More references should be cited, with this one (PMID: 40137747) as an example (citing is optional).
5. "There are also studies that have found a negative correlation between vitamin D and interleukin 6, which is a key cytokine in inflammatory storm induced by COVID-19 infection [25].": While the authors discuss an very important question on the clinical significance of IL-6 in COVID-19 cases, it is important to discuss that cytokine data should be interpreted carefully as cytokine detection tend to be affected by a diversity of biological and technical variables. More references should be cited, with this one (PMID: 33667962) as an example (citing is optional).

Please return the manuscript within 30 days; if you cannot complete the modification within this time period, please contact me. If you do not wish to modify the manuscript and prefer to submit it to another journal, notify me immediately so that the manuscript may be formally withdrawn from consideration by Spectrum.

Revision Guidelines

Publication Fees: For information on publication fees and which article types are subject to charges, visit our website. If your

manuscript is accepted for publication and any fees apply, you will be contacted separately about payment during the production process; please follow the instructions in that e-mail. Arrangements for payment must be made before your article is published.

Sincerely,
Benjamin Liu
Editor
Microbiology Spectrum

Reviewer #1 (Comments for the Author):

Peer Review Report

1. General Comments

This manuscript addresses an important topic-the impact of COVID-19 on pediatric patients-by analyzing the clinical characteristics and outcomes of children hospitalized in a tertiary hospital. The topic is timely and relevant, and the authors attempt to categorize patients by age group, which adds value. However, there are several issues that need to be addressed before the manuscript can be considered for publication.

2. Major Comments

1. Language and Grammar: The manuscript requires substantial language editing. Numerous grammatical and syntactical errors hinder readability. It is recommended that the authors seek professional English editing.
2. Study Design and Methodology: The retrospective design should be more clearly explained. The inclusion criteria for the patients are not fully described. Were all COVID-19 positive children admitted to the hospital? Or only those with severe symptoms?
3. Statistical Analysis: While the use of the Kruskal-Wallis test is mentioned, the rationale for choosing this test over others is not explained. Also, details of how the data met the assumptions for this test are missing.
4. Discussion Section: The discussion includes interesting points about ACE2 expression and gender differences, but it would benefit from more integration of the study findings with the existing literature.
5. Limitations: The authors do mention some limitations, but they should be elaborated. For example, the study is based on data from a single center and during a very specific timeframe (during lockdowns), which limits generalizability.

3. Minor Comments

6. Please revise the title for clarity.
7. Abbreviations such as MIS-C should be spelled out upon first use.
8. Some sentences are awkwardly constructed. For instance, "due to positive COVID-19" should be corrected to "due to testing positive for COVID-19."
9. Ensure consistent formatting of tables and figures.

4. Recommendation

Recommendation: Major Revision

Reviewer #3 (Comments for the Author):

1) The manuscript requires substantial English editing. Numerous spelling mistakes, grammatical errors, and awkward sentence constructions affect clarity and readability. Below are notable examples:

Spelling Errors:

Line 72: "INTRIDUCTION" should be "INTRODUCTION"

Line 97: "MATERIALS AND METHIDS" should be "MATERIALS AND METHODS"

Line 290: "CONCLUTION" should be "CONCLUSION"

Line 337: "ACKNOWLEDFEMENTS" should be "ACKNOWLEDGEMENTS"

Other Language Issues:

"New Corona" (multiple occurrences) should be replaced with "COVID-19" or "SARS-CoV-2" throughout for scientific accuracy.

2) The study design is descriptive. Although it adds local data, the novelty is limited unless the authors can better contextualize their findings with respect to international literature.

3) The impact of vaccination on outcomes is only briefly mentioned. A clearer breakdown of outcomes by vaccination status

would be useful.

4) The table is highly informative but difficult to read due to formatting. Please reformat for clarity.

5) There is limited information about the statistical methodology. The authors mention using Kruskal-Wallis H tests, but do not explain why this non-parametric test was selected or whether assumptions for statistical tests were met.

Reviewer Comments on Manuscript

Title: *Clinical characteristics and outcomes of COVID-19 for children*

This manuscript presents a retrospective study of 237 pediatric COVID-19 cases admitted to a hospital in Xiamen, China. It evaluates age-related symptom profiles, severity of disease, and outcomes, with an emphasis on identifying high-risk subgroups such as infants and males. The topic is timely and relevant, especially in light of the still-evolving understanding of pediatric SARS-CoV-2 infections.

Comments:

1) The manuscript requires substantial English editing. Numerous spelling mistakes, grammatical errors, and awkward sentence constructions affect clarity and readability. Below are notable examples:

Spelling Errors:

Line 72: "INTRIDUCTION" should be "INTRODUCTION"

Line 97: "MATERIALS AND METHIDS" should be "MATERIALS AND METHODS"

Line 290: "CONCLUTION" should be "CONCLUSION"

Line 337: "ACKNOWLEDFEMENTS" should be "ACKNOWLEDGEMENTS"

Other Language Issues:

"New Corona" (multiple occurrences) should be replaced with "COVID-19" or "SARS-CoV-2" throughout for scientific accuracy.

2) The study design is descriptive. Although it adds local data, the novelty is limited unless the authors can better contextualize their findings with respect to international literature.

3) The impact of vaccination on outcomes is only briefly mentioned. A clearer breakdown of outcomes by vaccination status would be useful.

4) The table is highly informative but difficult to read due to formatting. Please reformat for clarity.

5) There is limited information about the statistical methodology. The authors mention using Kruskal-Wallis H tests, but do not explain why this non-parametric test was selected or whether assumptions for statistical tests were met.

Response to Reviewers

Dear Prof. Benjamin Liu:

Thank you very much for your work with our manuscript. We accept all constructive suggestions of reviewers and respective corresponding corrections about manuscript.

Editor's comments 1: Omicron with key mutations in envelope gene is the direct reason of the mild clinical characteristics and outcomes of COVID. The authors should introduce or discuss this point. More references should be cited, with this one (PMID: 39744807) as an example (citing is optional).

Authors: Thank you for your suggestion. Now it is discussed in the manuscript.

Editor's comments 2: Introduction should include diagnostic methods for COVID. More references should be cited, with this one (PMID: 39857007) as an example (citing is optional).

Authors: Thank you for your suggestion, and it has now been cited in the article.

Editor's comments 3: "Adult infection with SARS-CoV-2 is a highly inflammatory response characterized by acute respiratory distress syndrome (ARDS) as the main symptom, but

symptoms are mild in children": There are no references to support this statement. More references should be cited, with this one (PMID: 33337932) as an example (citing is optional).

Authors: Thank you. This is a good suggestion. Now it is discussed in the manuscript.

Editor's comments 4: "The decrease in the incidence rate of febrile convulsions may be caused by the decrease in the incidence rate of influenza, chickenpox and other infectious diseases known to lead to febrile convulsions due to social distance, wearing masks and hand hygiene". The authors discuss the interplay between COVID and flu. But COVID also has influence on flu epidemiology. The authors should discuss this. More references should be cited, with this one (PMID: 40137747) as an example (citing is optional).

Authors: Thank you for your suggestion. Now it has been revised.

Editor's comments 5: "There are also studies that have found a negative correlation between vitamin D and interleukin 6, which is a key cytokine in inflammatory storm induced by COVID-19 infection [25].": While the authors discuss an very important question on the clinical significance of IL-6 in COVID-19 cases, it is important to discuss that cytokine data should be interpreted carefully as cytokine detection tend to be affected by a diversity of biological and technical variables. More references should be cited, with this one (PMID: 33667962) as an example (citing is optional).

Authors: Thank you. Now it is discussed in the manuscript.

Reviewer 1 comments 1 : Language and Grammar: The manuscript requires substantial language editing. Numerous grammatical and syntactical errors hinder readability. It is recommended that the authors seek professional English editing.

Authors: I greatly appreciate your suggestions. We have had a professional English editor assist with the revisions.

Reviewer 1 comments 2 : Study Design and Methodology: The retrospective design should be more clearly explained. The inclusion criteria for the patients are not fully described. Were all COVID-19 positive children admitted to the hospital? Or only those with severe symptoms?

Authors: Thank you for the expression of your opinion. Inclusion criteria: children under 18 years old with confirmed COVID-19 admitted to the First Affiliated Hospital of Xiamen University. Now it is discussed in the manuscript.

Reviewer 1 comments 3 : Statistical Analysis: While the use of the Kruskal-Wallis test is mentioned, the rationale for choosing this test over others is not explained. Also, details of how the data met the assumptions for this test are missing.

Authors: Thank you for your suggestion. In this study, children were divided into four age grades according to their age, and differences between grades were compared, and each observation was independent of each other. Therefore, the Kruskal-wallis H test was used for statistical methods.

Reviewer 1 comments 4 : Discussion Section: The discussion includes interesting points about ACE2 expression and gender differences, but it would benefit from more integration of the study findings with the existing literature.

Authors: Thank you for your suggestion. Now it has been revised.

Reviewer 1 comments 5 : Limitations: The authors do mention some limitations, but they should be elaborated. For example, the study is based on data from a single center and during a very specific timeframe (during lockdowns), which limits generalizability.

Authors: Thank you for the expression of your opinion. Now it is discussed in the manuscript.

Reviewer 1 comments 6 : Please revise the title for clarity.

Authors: Thank you for the expression of your opinion. Now I've changed the title and

made it clear.

Reviewer 1 comments 7 : Abbreviations such as MIS-C should be spelled out upon first use.

Authors: Thank you for the expression of your opinion. I have now spelled out the manuscript when it was first used.

Reviewer 1 comments 8 : Some sentences are awkwardly constructed. For instance, "due to positive COVID-19" should be corrected to "due to testing positive for COVID-19."

Authors: Thank you for the expression of your opinion. Now I have modified the similar sentence structure in the manuscript.

Reviewer 1 comments 9 : Ensure consistent formatting of tables and figures.

Authors: Thank you for the expression of your opinion. Now I have revised it in the manuscript to ensure the format of tables and figures is consistent.

Reviewer 3 comments 1: The manuscript requires substantial English editing. Numerous spelling mistakes, grammatical errors, and awkward sentence constructions affect clarity and readability.

Authors: Thank you for your suggestion. Now it is corrected.

Reviewer 3 comments 2: The study design is descriptive. Although it adds local data, the novelty is limited unless the authors can better contextualize their findings with respect to international literature.

Authors: Thank you for your feedback. We have now more closely contextualized our findings with international research in the discussion section, supplemented with additional references. These revisions can be found in the discussion portion of the manuscript.

Reviewer 3 comments 3: The impact of vaccination on outcomes is only briefly mentioned. A clearer breakdown of outcomes by vaccination status would be useful.

Authors: Thank you for the expression of your opinion. Now I have presented the specific situation of children's vaccination in the table, you can see it in the attached table.

Reviewer 3 comments 4: The table is highly informative but difficult to read due to formatting. Please reformat for clarity.

Authors: Thank you for your feedback. We have adjusted the table format, and the revised

version can be found in the attached table.

Reviewer 3 comments 5: There is limited information about the statistical methodology.

The authors mention using Kruskal-Wallis H tests, but do not explain why this non-parametric test was selected or whether assumptions for statistical tests were met.

Authors: Thank you for your suggestion. In this study, children were divided into four age grades according to their age, and differences between grades were compared, and each observation was independent of each other. Therefore, the Kruskal-wallis H test was used for statistical methods.

Cordially thanks for your helpful suggestions.

From authors,

Yafang Hong

Re: Spectrum01577-25R1 (**Pediatric COVID-19: Clinical characteristics and prognostic outcomes**)

Dear Ms. Yafang Hong:

Your manuscript has been accepted, and I am forwarding it to the ASM production staff for publication. Your paper will first be checked to make sure all elements meet the technical requirements. ASM staff will contact you if anything needs to be revised before copyediting and production can begin. Otherwise, you will be notified when your proofs are ready to be viewed.

Sincerely,
Benjamin Liu
Editor
Microbiology Spectrum